# The Changing Landscape of Thyroid Surgery during the COVID-19 Pandemic: A Four-Year Analysis in a University Hospital in Romania

**DOI:** 10.3390/cancers15113032

**Published:** 2023-06-02

**Authors:** Catalin Vladut Ionut Feier, Calin Muntean, Alaviana Monique Faur, Andiana Blidari, Oana Elena Contes, Diana Raluca Streinu, Sorin Olariu

**Affiliations:** 1First Discipline of Surgery, Department X-Surgery, “Victor Babes” University of Medicine and Pharmacy, 2 E. Murgu Sq., 300041 Timisoara, Romania; catalin.feier@umft.ro (C.V.I.F.); olariu.sorin@umft.ro (S.O.); 2First Surgery Clinic, “Pius Brinzeu” Clinical Emergency Hospital, 300723 Timisoara, Romania; diana.streinu@umft.ro; 3Medical Informatics and Biostatistics, Department III-Functional Sciences, “Victor Babes” University of Medicine and Pharmacy, 2 E. Murgu Sq., 300041 Timisoara, Romania; 4Faculty of Medicine, “Victor Babes” University of Medicine and Pharmacy, 300041 Timisoara, Romania; alaviana.faur@student.umft.ro; 5Oncology, Department IX-Surgery, “Victor Babes” University of Medicine and Pharmacy, 2 E. Murgu Sq., 300041 Timisoara, Romania; andiana.blidari@umft.ro (A.B.); oana.conte@umft.ro (O.E.C.)

**Keywords:** thyroid surgery, COVID-19 pandemic, surgery duration, length of hospitalization

## Abstract

**Simple Summary:**

The global impact of the COVID-19 pandemic has led to significant changes in healthcare systems worldwide, including the prioritization of infectious disease management and the postponement of elective surgery. Furthermore, this analysis revealed an increase in poor prognostic factors associated with thyroid cancer following the onset of COVID-19. Notably, there has been a recent increase in the number of thyroid surgeries performed, along with a surge in the incidence of malignant thyroid tumors. These findings suggest potential implications for disease progression and treatment outcomes in patients with thyroid cancer during the pandemic.

**Abstract:**

The aim of this study was to highlight the changes in the surgical treatment of patients with thyroid pathology over a 4-year period. The dynamics of various parameters during this period at a tertiary University Hospital in Timisoara, Romania were examined. Data from 1339 patients who underwent thyroid surgery between 26 February 2019 and 25 February 2023 were analyzed. The patients were divided into four groups: Pre-COVID-19, C1 (first year of the pandemic), C2 (second year), and C3 (third year). Multiple parameters of the patients were analyzed. Statistical analysis revealed a significant decrease in the number of surgical interventions performed during the first two years of the pandemic (*p* < 0.001), followed by an increase in subsequent periods (C3). Furthermore, an increase in the size of follicular tumors was observed during this period (*p* < 0.001), along with an increase in the proportion of patients with T3 and T4 stage in C3. There was also a reduction in the total duration of hospitalization, postoperative hospitalization, and preoperative hospitalization (*p* < 0.001). Additionally, there was an increase in the duration of the surgical procedure compared to the pre-pandemic period (*p* < 0.001). Moreover, correlations were observed between the duration of hospitalization and the duration of the surgical procedure (r = 0.147, *p* < 0.001), and between the duration of the surgical procedure and postoperative hospitalization (r = 0.223, *p* < 0.001). These findings confirm the modification of clinical and therapeutic management of patients who underwent thyroid surgery over the past 4 years, with the pandemic generating an impact whose full consequences are not yet fully known.

## 1. Introduction

The global healthcare systems have faced significant challenges due to the COVID-19 pandemic [1]. Originating in the province of Wuhan, China, at the end of 2019, the COVID-19 outbreak has rapidly spread, requiring crucial modifications in treatment protocols, particularly for patients undergoing elective surgeries [2,3,4]. The implementation of epidemiological measures and government-imposed restrictions, combined with patient apprehension regarding potential viral transmission, has resulted in a substantial decrease in endocrinology clinic visits throughout this period. Consequently, there has been a notable decline in the number of thyroid surgeries, despite thyroid pathology being one of the most prevalent endocrine disorders, with over 80% of cases being incidentally detected [5].

Thyroid cancer, as one of the most frequently encountered endocrine malignancies, exhibits a comparatively favorable prognosis compared to other forms of cancer. Notably, Globocan reported 449,000 newly diagnosed cases of thyroid cancer in females and 137,000 cases in males in 2020 [6]. In light of the pandemic, certain guidelines have recommended the deferral of surgical interventions. However, delaying surgical procedures may lead to the progression of capsular and vascular invasions, including lymph node metastasis. Regrettably, limited studies have comprehensively evaluated the management outcomes of patients with thyroid pathology necessitating surgical intervention during the pandemic [7].

To address this knowledge gap, the present study aims to provide a comprehensive analysis and interpretation of the clinical and pathological aspects of patients undergoing thyroid surgery within this unique period. By tracking the dynamic changes in various parameters over the past four years, our investigation seeks to elucidate the impact of the pandemic on patients undergoing thyroid surgery at the “Pius Brinzeu” Emergency County Clinical University Hospital in Timisoara, Romania.

## 2. Materials and Methods

The Pius Brinzeu Emergency County Hospital in Timisoara comprises three departments of General Surgery where thyroid surgery is performed. The study analyzed data using the institution’s database from 1339 patients who underwent thyroid surgery between 26 February 2019 and 25 February 2023. It is worth noting that the first confirmed case of COVID-19 in Romania occurred on 26 February 2020, with the emergency state being introduced on 16 March 2020. To provide a better perspective, the study period was divided into four groups:“Pre-COVID-19” (from 26 February 2019 to 25 February 2020);“C1” (COVID-19-first year, from 26 February 2020 to 25 February 2021);“C2” (COVID-19-s year, from 26 February 2021 to 25 February 2022);“C3” (COVID-19-third year, from 26 February 2022 to 25 February 2023).

Data on the monthly number of COVID-19 cases in Romania over the past three years were obtained from the open statistical data provided by The National Institute of Public Health (https://insp.gov.ro/ (accessed on 22 April 2023)) [8].

As this hospital is a tertiary institution where an important number of thyroid surgeries are performed, all patients included in the study were referred to the investigated institution by primary or secondary care providers.

The study included only patients who underwent thyroid surgery during the aforementioned periods. Additional inclusion and exclusion criteria were implemented during the pandemic. Inclusion criteria required the absence of any symptoms related to SARS-CoV-2 infection at the time of admission or within 7 days prior, with patients undergoing epidemiological screening upon entry into the institution. Furthermore, patients had to test negative for SARS-CoV-2 using the RT-PCR test upon admission to the surgical clinics. Exclusion criteria during the pandemic included the presence of any COVID-19-specific symptoms at the time of admission or within 7 days prior, a positive RT-PCR test result, or the development of a SARS-CoV-2 infection during hospitalization. The aim of this study was to present the impact of the COVID-19 pandemic on these patients, and not the direct impact of the COVID-19 disease on their health and evolution.

Multiple parameters were investigated, including age, gender, environment origin, total duration of hospitalization, as well as preoperative and postoperative hospitalization durations. Other factors considered were the type of surgical intervention (total thyroidectomy, lobectomy), duration of the surgical procedure, and the Charlson comorbidity index. Following the surgery, the excised thyroid lobe/lobes were sent for morphopathological examination. Hence, histopathological type, tumor size, presence or absence of multifocality, vascular and capsular invasion, as well as T staging (tumor invasion), the number of lymph nodes with metastases (N), and the presence or absence of metastases (M) were taken into account. A more extensive analysis was performed specifically on papillary carcinoma and follicular tumors.

For data analysis and interpretation, the IBM SPSS Statistics program for Windows (IBM, Armonk, NY, USA) was utilized. Central tendency and dispersion parameters were determined for numerical variables. Categorical variables were analyzed using frequency tables and percentages. The analysis of variance (ANOVA) test was used to identify differences between study groups for continuous variables, while the chi-square test was employed to determine differences in proportions. Pearson and Spearman correlation coefficients were used to highlight correlations between study variables. Statistical significance was considered at a *p*-value of <0.05.

Data collection was conducted after obtaining ethical approval from the Hospital Commission (No. 391/20.04.2023).

## 3. Results

From 26 February 2020 to 25 February 2023, a total of 3,340,342 cases of COVID-19 were recorded in Romania, and 882 patients underwent surgical procedures during this period. The monthly M (average) of COVID-19 cases during this period was 92,787.28 cases (min—2242, max—516,959), with a standard deviation (SD) of 124,350.16. The monthly average of thyroid surgeries performed was 24.50 (min—2, max—50), with a standard deviation (SD) of 13.41.

To investigate the correlation between these two variables over the three years of the pandemic, a statistical analysis was performed. The Pearson correlation coefficient was calculated, resulting in r = −0.246, with a *p*-value of 0.149, indicating that the correlation was negative but not statistically significant.

To provide further insights, the correlation between the two variables was examined individually each year of the pandemic: C1, C2, and C3. Since the monthly average of COVID-19 cases and the number of surgical procedures were considered, and there were 12 months in each period (N < 20), a non-parametric Spearman’s correlation coefficient (rho) was utilized for the individual correlations.

Table 1 presents the variation of the Spearman coefficient and the average monthly number of COVID-19 cases, alongside the average number of surgical procedures performed.

Thus, for the C1 and C3 periods, a weak positive correlation was observed (0.035, 0.049), but it was not statistically significant (*p* > 0.05). In contrast, for the C2 period, a negative correlation was found (−0.314), but it was also not statistically significant (*p* > 0.05).

Figure 1 illustrates the monthly variation of COVID-19 cases and the monthly variation of the number of surgical procedures performed over the course of the three years.

Table 2 presents the clinical data of patients who underwent thyroid surgery at the University Hospital in Timisoara, Romania.

Out of the 1339 patients, 34.1% underwent surgery in the Pre-COVID-19 period, 13.3% in the C1 period, 20.1% in the C2 period, and 32.5% in the C3 period. The application of the chi-square test yielded a *p*-value of less than 0.05, indicating a statistically significant difference among the four periods.

The longest hospitalization duration was reported in the Pre-COVID-19 period, with statistically significant differences observed compared to the C2 and C3 periods (*p* < 0.001). Furthermore, significant differences were found between the first year of the pandemic and C2 and C3 periods (*p* = 0.003), with a decrease in average duration from 4.63 ± 2.06 days to 4.03 ± 2.27 days and 4.08 ± 1.79 days, respectively.

Regarding preoperative hospitalization, statistically significant differences were observed between the pre-pandemic period and C2 and C3 periods (*p* < 0.001), as well as between the first year of the pandemic and the subsequent periods (*p* < 0.001).

The average duration of postoperative hospitalization decreased over time, from 2.98 ± 1.57 days in the Pre-COVID-19 period to 2.60 ± 1.20 days in the C3 period. Significant differences were observed between the Pre-COVID-19 period and the C2 and C3 periods (*p* < 0.001).

There was a correlation between age and duration of hospitalization (r = 0.158, *p* < 0.001), a correlation between duration of hospitalization and surgical intervention duration (r = 0.147, *p* < 0.001), and a correlation between surgical intervention duration and postoperative hospitalization duration (r = 0.223, *p* < 0.001).

The most common diagnosis was a follicular tumor, specifically benign nodules, which was found in 63% of the patients based on histopathological examination. The characteristics of these patients are presented in Table 3.

In addition to the significant differences observed in the entire patient cohort, there was a statistically significant variation in the average size of nodules across the four study periods. The minimum average nodule size was observed in the Pre-COVID-19 period (1.65 ± 0.91 cm), while the maximum average size was seen in the first year of the pandemic (2.20 ± 1.72 cm).

Throughout the four years, 22 patients with follicular cancer were reported. A total of 18.18% (4 cases) occurred in the Pre-COVID-19 period, 18.18% (4 cases) in the C1 period, 22.72% (5 cases) in the C2 period, and 40.9% (9 cases) in the C3 period, indicating significant differences among the four periods (*p* < 0.05).

Regarding (WDT-UMP), 29 cases were reported over the four periods, with 24.13% (7 cases) in the Pre-COVID-19 period, 10.34% (3 cases) in the C1 period, 20.69% (6 cases) in the C2 period, and 44.82% (13 cases) in the C3 period.

For (NIFTP), 11 cases were reported during the four-year period, with 54.54% (6 cases) in the Pre-COVID-19 period and the remaining 45.45% (5 cases) in the C3 period.

The sizes of follicular tumor nodules differed significantly across the four periods (*p* < 0.001). The average sizes were 1.66 cm in the Pre-COVID-19 period, 2.16 cm in the C1 period, 1.7 cm in the C2 period, and 1.95 cm in the last period.

The most common type of thyroid cancer was papillary carcinoma, with 26.7% of patients diagnosed based on histopathological examination. The characteristics of these patients and their variation across the four periods are presented in Table 4.

Significant differences were observed in the total duration of hospitalization, preoperative hospitalization duration, and postoperative hospitalization duration (*p* < 0.001) across the four periods, with a progressive decrease in these values.

Furthermore, an increase in the average duration of the surgical intervention was noted from 133.15 ± 46.85 min (Pre-COVID-19) to 160.72 ± 71.21 min in the C3 period.

In patients with papillary carcinoma, analyzing the mean Charlson Index value among patients between periods revealed a *p*-value of 0.05 between the Pre-COVID-19 period and C2. Patients with benign lesions had a mean Charlson Index of M = 3 in the Pre-COVID-19 period and M = 2.73 in the C1 period, with a *p*-value of 0.04 obtained between the two periods.

When considering the gender of patients, a higher average duration of surgical intervention was observed in males (M = 161.68 min vs. M = 145.94 min), with a *p*-value of 0.005. Additionally, males had a higher average Charlson Index value (M = 3.18 vs. M = 2.97), with a *p*-value of 0.043. The same trend was observed in terms of tumor size (M = 1.8 cm for males vs. M = 1.44 cm for females, *p* = 0.022), and vascular invasion was significantly more common in males (14.3% vs. 7.8%, *p* = 0.006).

## 4. Discussion

During the COVID-19 pandemic, there was a significant impact on healthcare systems worldwide, including the clinical and therapeutic management of patients undergoing elective surgery. Patients with thyroid pathology requiring surgical intervention were not an exception to this situation. Due to insufficient data about the SARS-CoV-2 virus and the high transmission rate and aggressiveness of this disease, especially in patients with comorbidities, the main human and medical resources were redirected to support COVID-19 patients, while the rest of the pathologies were initially slightly neglected [9,10,11,12,13]. Moreover, constraints mandated by governing bodies, coupled with their recommendation to solely seek medical care for emergencies, patients’ apprehension towards venturing outside and potentially encountering individuals, as well as their reluctance to seek treatment at medical facilities due to the heightened susceptibility to infection, precipitated a substantial decline in the quantity of surgical procedures conducted during this timeframe [14,15,16].

International guidelines have suggested the temporary postponement of surgical interventions in the case of cancers, given this would not have a significant impact on patients’ prognosis [17,18,19]. However, there are few studies that present the exact impact of these delays on the evolution of the pathology. In a study conducted by clinicians specialized in thyroid diseases, 87.9% were concerned about the delay in treatment for their patients during the pandemic [20]. For this reason, further research is needed on how thyroid cancer is treated after COVID-19.

This study spanned 4 years, with the aim of monitoring the changing landscape of thyroid surgery. During this study, a dramatic decrease in the number of surgical interventions performed in the first year of the pandemic compared to the previous period was observed. There was a 61% decrease in the number of thyroid surgeries performed in our hospital in the first year of the pandemic. In period C2, this number decreased by 41.13% compared to the Pre-COVID-19 period, and in the fourth year, a difference of 4.81% was noted. It can be concluded that, in terms of the number of surgical interventions, it has “returned” to pre-pandemic levels. Thus, patients who were postponed or for various reasons did not undergo endocrinological monitoring were able to return to the hospital to receive treatment once the aggressiveness of the pandemic was reduced. There was a decrease of 27.1% in the first year in Italy, and a study from Greece presented a decrease of 64.8% in the first phase, followed by a decrease of 44.7% in the second phase and 5.1% in the third phase of the pandemic [7,21]. This variation is justified by the severity of restrictions imposed by authorities at the beginning of the pandemic. However, with the passage of time, the severity of restrictions has decreased, patients’ fears have reduced, and vaccination against COVID-19 has begun, leading to a gradual return to normal surgical activity, with the number of surgeries performed slowly returning to pre-pandemic levels.

The results obtained show an inverse correlation between the average monthly number of COVID-19 cases and the average monthly number of surgical interventions performed in Romania (r = −0.246, with a *p*-value of 0.149). This correlation, although statistically insignificant, cannot be neglected, as it is justified by the application of more severe epidemiological measures during pandemic waves on both the population and hospitals. Patients’ reluctance to visit hospitals during times with a high incidence of COVID-19 cases should also be considered. A study conducted in South Korea showed a weak and statistically non-significant negative correlation (r = −0.170, *p* = 0.598) between the pre-pandemic period and the first year of the pandemic, indicating a slight decrease in surgical activity during that time. However, during the second year of the study, a positive correlation was observed between the number of COVID-19 cases and the number of surgical procedures performed (r = 0.668, *p* = 0.017), suggesting a recovery of surgical activity in the institution [22].

The surgical society guidelines worldwide have created directives to guide the therapeutic management of patients with thyroid disease during the pandemic, with one indication being the reduction in patient exposure to the hospital environment and consequently reducing the risk of contacting the novel coronavirus [23,24]. In this study, to reduce the risk of virus transmission within the hospital, patients followed a well-established protocol set by the hospital’s management. Patients underwent epidemiological triage upon entering the hospital and were tested for COVID-19 with RT-PCR to ensure they were negative. They were isolated in the first 24 h from admission, and this was applied until the end of the C2 period. Medical staff aware of the epidemic risk tried to reduce patient contact with the hospital environment. As a result, there were significant differences in the total duration of hospitalization, preoperative hospitalization duration, postoperative hospitalization duration, and surgical intervention duration between the four periods, both in the general population and in patients with papillary cancer or benign lesions.

Regarding the duration of preoperative hospitalization, it was highest during the initial phase of the pandemic. It should be noted that this value was influenced by the 24 h isolation period until the RT-PCR test results were obtained. Furthermore, the number of surgical interventions performed at the hospital was reduced due to stringent epidemic control measures, including longer sterilization times for the operating room. Finally, in order to prevent postoperative complications and shorten the postoperative hospital stay, patients underwent additional preoperative investigations and were carefully managed to maintain a balance in their electrolyte levels, as well as their hemodynamics, which aided in their postoperative recovery.

In the following periods, C2 and C3, patients had a reduced preoperative hospitalization period from an average of 1.9 days to 1.45 ± 0.99 and 1.48 ± 1.26 days, respectively. This is due to the relaxation of restrictions, and the possibility of patients undergoing endocrinological outpatient visits and preoperative investigations, thus reducing the time spent in hospital before surgery. Moreover, in the last period of this study, patients were no longer required to undergo an RT-PCR test upon admission to the University Hospital if they provided evidence of anti-COVID-19 vaccination. It should be noted that the average duration of hospitalization was similar to the period when RT-PCR testing was mandatory. However, when making this comparison, it must be taken into consideration that due to the increased number of surgical interventions performed, which was almost double that of the previous period, a larger number of patients also meant a longer waiting time. The differences in this regard between the Pre-COVID-19 period (1.78 ± 1.48) and C2–C3 (1.45 ± 0.99, and 1.48 ± 1.26 days) and between the C1 period (1.90 ± 1.36) and the C2–C3 periods (1.45 ± 0.99 and 1.48 ± 1.26 days) are statistically significant.

As for the postoperative hospitalization period, there was a significant decrease in its average value, *p* < 0.01. The longest average postoperative hospitalization period occurred in the Pre-COVID-19 period with 2.98 ± 1.57 days, followed by a progressive decrease over the three pandemic periods. Surgeons preferred to shorten the hospitalization period to reduce the risk of patients contacting the novel coronavirus. Thus, when patients were hemodynamically stable and did not present postoperative complications, they were discharged. This practice was also observed in a hospital in South Korea, where a significant decrease in the length of hospital stay of patients after surgery was observed [22]. Although the incidence of COVID-19 cases decreased significantly in the C3 period, this practice of reducing the postoperative hospitalization period remained valid. It should be noted that due to the large number of surgical interventions performed in the last period, it is necessary to shorten this period in order to surgically treat as many patients as possible and not delay the timing of the surgery further.

Global surgical delays could be devastating, as reported in a study of 3,672,561 cancer patients, which showed that the time to treatment initiation was directly associated with the risk of mortality from 1.2% to 3.2% in early-stage cancers [25]. Therefore, it is important to strike a balance between maintaining safety measures to prevent the spread of COVID-19 and providing timely surgical care for patients in need.

In terms of surgical duration, there was an increase in the mean duration of surgery over the four periods, from a mean of 133.19 ± 45.66 min in the Pre-COVID-19 period to 161.63 ± 61.76 min in period C3. In the case of benign tumors, this prolonged surgery duration may be justified by a significant increase in the size of benign nodules (1.65 ± 0.91 cm to 1.85 ± 0.96 cm, with a *p* < 0.001 between the first and last period). Current studies demonstrate that benign nodules exhibit slow, progressive growth, limited to approximately 5 mm in 5 years for the main nodule in cases of multinodular disease [26].

The proportion of patients who underwent a total thyroidectomy (TT) for differentiated thyroid cancer (TC) is significantly higher than the ones with a subtotal thyroidectomy (STT). According to the available literature, there is ongoing controversy regarding the optimal surgical treatment between the two. The debate mainly arises when considering young patients with small tumors confined to one lobe of the thyroid. Studies have shown that a total thyroidectomy with TSH suppression significantly reduces recurrence rates and enables accurate staging, while also facilitating subsequent radioactive iodine ablation [27].

Furthermore, a study involving 306 patients reported that postoperative hematoma occurred in 1.78% of cases in the TT group and 2.17% of cases in the STT group, with no statistically significant differences between the two groups. It is worth noting that temporary recurrent laryngeal nerve palsy occurred in 15.89% of patients undergoing TT and only 11.96% of patients undergoing STT, but these differences were not statistically significant [28].

The number of patients with a histopathological diagnosis of benign nodules varied by 18% between the two periods at the extremes (Pre-COVID-19: 307 cases, C3: 253 cases). However, there was a 63.51% decrease in the number of cases in period C1 compared to Pre-COVID-19, and a 43.97% decrease compared to period C2. The results of this study show how the number of cases started to increase alongside the progression of the pandemic. Thus, in the last year, 225.89% more benign nodules were identified compared to period C1 and 147.09% more nodules compared to period C2. This increase in the number of surgeries performed and the number of patients diagnosed with benign nodules is due to patients who postponed their visit to the specialist for various reasons previously presented, but also an increase in the screening of thyroid pathology.

Therefore, patients have started to rely more confidently on medical services compared to the early stages of the pandemic. It is worth noting the evolution of the average size of benign nodules. The statistical tests reveal significant differences between the four periods. Thus, in period C1, the highest average size of 2.20 ± 1.72 was identified, supported by the fact that only patients with larger nodules with associated symptoms were referred to the surgical departments. However, there was a decrease in this mean value in period C2, but an increase to 1.85 ± 0.96 cm in period C3.

Although our study has its limitations, as there are no studies to support this aspect, there are a few mentions to be made. During the pre-COVID period, the average size of thyroid nodules was 1.65 ± 0.91 cm, and the number of patients was similar to that in period C3. Therefore, we can speculate that the first two years of the pandemic, with delayed surgical intervention, decreased screening, and fewer diagnostic tests performed, have likely contributed to the progression of thyroid pathology. Studies have shown a significant decrease in the reported use of ultrasounds in the office, fine needle aspiration, and laryngoscopy during this period, all with *p* < 0.001 [29].

The results of thyroid fine needle aspiration (FNA) are classified and reported using The Bethesda System for Reporting Thyroid Cytopathology (TBSRTC). Although this study has its limitations and does not provide data on this aspect, it is worth noting that this system is internationally used in the management of patients with thyroid nodular pathology. The TBSRTC comprises six categories, and the risk of incidental malignancy increases from category I to category VI [30]. In the literature, information regarding category III varies. While the risk of incidental malignancy in category II is generally low, ranging from 0% to 3%, it varies between 5% and 15% in category III, with some studies reporting percentages as high as 26.6% to 37.8%. As a result, these patients may have a higher risk of malignancy than traditionally believed [31,32].

Throughout these years, the size of benign nodules in patients has increased, which is a cause for concern. A study in South Korea also shows a significant increase in the size of follicular tumors (including benign nodules) between the period before COVID-19 and 1 year and 2 years after the onset of the pandemic (3.5 ± 2.2, 4.0 ± 1.9, 4.3 ± 2.3 cm, *p* = 0.022) [22].

The proportion of patients presenting with papillary carcinoma measuring less than 1 cm remains constant throughout the four periods (ranging from 64.6% to 70.25% in the Pre-COVID-19 period). However, the number of patients presenting with papillary cancer is higher in period C3 by 14.38% compared to the Pre-COVID-19 period, almost triple compared to period C1, and almost double compared to period C2. Thus, significant statistical differences have been highlighted between the four periods. This proportion of papillary microcarcinoma and its variation during the pandemic has been presented in other studies (with a variation ranging from 65.5% to 69.8%), but without a significant difference in the number of patients throughout the studied periods [22]. Adherence to these characteristics in patients from other parts of the world demonstrates that well-differentiated papillary thyroid cancers have not been significantly affected by surgical delays. Therefore, in the context of the COVID-19 pandemic, surgical delays for most well-differentiated thyroid cancers are unlikely to have a significant impact on patient outcomes [33].

Taking into account that during the pandemic, only severe cases were being treated, and patients were only visiting hospitals or specialized clinics in critical situations, we can observe how the proportions of patients with papillary cancer have increased in terms of capsular invasion (54.4% compared to 43.2%) and vascular invasion (30.4% compared to 21.6%). The last period of this study showed a slight decrease in the proportion of patients with these characteristics, but an increase in those presenting with Stage T3 or T4 (9%-C3 compared to 3.7%-C1 or 2.5%-C2), given that the number of these patients doubled in period C3. Although the differences are not statistically significant, the evolution of these prognostic factors should be noted, as they have an impact on patient prognosis. In this regard, a study from Korea showed a significant association between the pandemic period and an increase in the proportion of poor prognostic factors (extrathyroidal extension, lymphatic invasion, vascular invasion, and lymph node metastasis in the neck, such as N1a and N1b) [21]. A study from China supports these findings and states that negative prognostic factors (previously mentioned) showed a higher proportion after the onset of the pandemic [34]. Similarly, a study conducted in Greece showed an increase in the proportion of patients with capsular invasion from 31.2% to 36.5% during the pandemic, as well as those with lymph node metastases, from 37% to 45% [21].

The statistical analysis revealed a correlation between age and length of hospitalization (r = 0.158, *p* < 0.001), indicating that older patients required longer surgical interventions. Prolonged surgical intervention also resulted in longer hospital stays (r = 0.147, *p* < 0.001), particularly in postoperative hospitalization (r = 0.223, *p* < 0.001). Prolonged surgical intervention may be attributed to a larger nodule size, or advanced T or N stage, which requires more attention and difficulty in performing the surgical intervention.

The limitations of this study include being conducted in a single university tertiary hospital in Romania. Although significant changes have been identified in the clinical and therapeutic management of patients with thyroid pathology, this study cannot generalize the situation nationally. Moreover, only patients who did not present with a COVID-19 infection prior to or during hospitalization were considered. The aim of this study was to present the way in which the pandemic has affected the surgical management of these patients, and not the impact of SARS-CoV-2 infection on these patients. Nonetheless, the strength of this study lies in describing the situation in 2022–2023, which shows a slight return to normal management, with consequences whose effects are not fully known.

## 5. Conclusions

The COVID-19 pandemic has had a significant impact on elective surgical activity. Thyroid pathology is a common condition, and its management has been influenced by global events over the past three years. Although the proportions of histopathological tumor types have remained constant over the four-year period, there is an observed increase in the number of patients with cancer in the last year, with reported differences in the morphopathological characteristics of the tumor. Careful management of these patients in the following period is necessary to ensure surgical treatment for both those who have postponed their interventions and those who have recently been referred to surgical services.

## Figures and Tables

**Figure 1 cancers-15-03032-f001:**
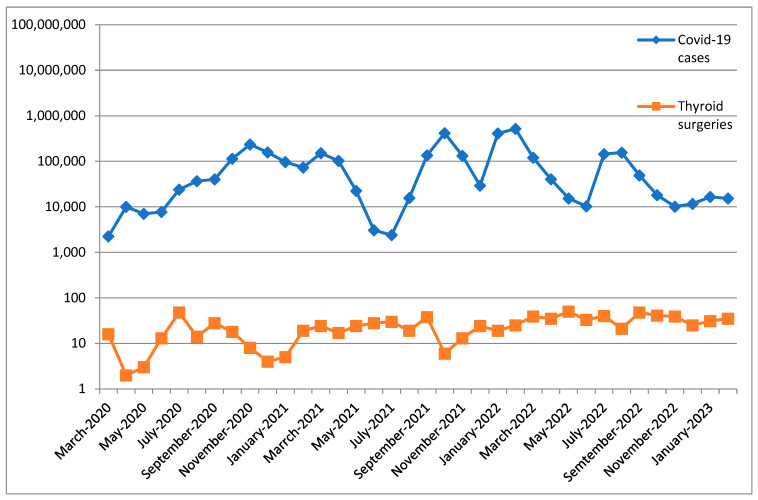
Variation of COVID-19 cases and number of thyroid surgeries performed.

**Table 1 cancers-15-03032-t001:** The monthly average COVID-19 cases and thyroid surgeries performed.

Period	COVID-19 Cases	Thyroid Surgery	Spearman’s Rho	*p*
C1 (M ± SD)	66,832.58 ± 72,069.597	14.83 ± 13.044	0.035	0.914
C2 (M ± SD)	160,957.75 ± 182,060.316	22.25 ± 8.292	−0.314	0.320
C3 (M ± SD)	50,571.50 ± 55,781.587	36.42 ± 8.415	0.049	0.879

**Table 2 cancers-15-03032-t002:** Clinic characteristics of patients.

Variables	Pre-COVID-19 N = 457	C1 N = 178	C2 N = 269	C3 N = 435	*p*
Age (years. M ± SD)	52.88 ± 13.31	52.69 ± 13.56	53.38 ± 13.75	54.58 ± 12.91	0.206
Gender					0.425
Male	48 (10.5%)	26 (14.6%)	30 (11.2%)	57 (13.1%)	
Female	409 (89.5%)	152 (85.4%)	239 (88.8%)	378 (86.9%)	
Environment					0.696
Urban	286 (62.6%)	116 (65.2%)	171 (63.6%)	263 (60.5%)	
Rural	171 (37.4%	62 (34.8%)	98 (36.4%)	172 (39.5%)	
Hospital stay (days. M ± SD)	4.77 ± 2.34	4.63 ± 2.06	4.03 ± 2.27	4.08 ± 1.79	<0.001 *
Preoperative stay (days. M ± SD)	1.78 ± 1.48	1.90 ± 1.36	1.45 ± 0.99	1.48 ± 1.26	<0.001 *
Postoperative stay (days. M ± SD)	2.98 ± 1.57	2.73 ± 1.31	2.58 ± 1.93	2.60 ± 1.20	<0.001 *
Duration of surgery (minutes. M ± SD)	133.19 ± 45.66	140.02 ± 52.70	155.58 ± 61.19	161.63 ± 61.76	<0.001 *
Type of surgery					0.770
Total Thyroidectomy	372 (81.4%)	149 (83.7%)	226 (84.0%)	361 (83.0%)	
Lobectomy	84 (18.4%)	28 (15.7%)	42 (15.6%)	74 (17.0%)	
Completion surgery after initial lobectomy	11 (12.9%)	6 (21.4%)	13 (31.0%)	18 (24.7%)	0.093
Charlson index (M ± SD)	3.11 ± 1.41	2.87 ± 1.14	2.95 ± 1.18	2.96 ± 1.12	0.099
Pathologic Diagnosis					0.140
Anaplastic cancer	1 (0.2%)	3 (1.7%)	0 (0%)	2 (0.5%)	
Medullary cancer	2 (0.4%)	1 (0.6%)	4 (1.5%)	7 (1.6%)	
Papillary cancer	125 (27.5%)	54 (30.3%)	79 (29.6%)	146 (33.6%)	
Follicular tumors **	324 (70.9%)	119 (66.8%)	184 (68.5%)	280 (64.4%)	
Lymphoma	1 (0.2%)	0 (0%)	1 (0.4%)	0 (0%)	

* *p* value lower than 0.05; ** Benign nodule, noninvasive follicular thyroid neoplasm with papillary-like nuclear features (NIFTP), follicular thyroid carcinoma, and well-differentiated thyroid tumor of uncertain malignant potential (WDT-UMP).

**Table 3 cancers-15-03032-t003:** Clinical and pathological Characteristic of patients with benign nodule.

Variables	Pre-COVID-19 N = 307	C1 N = 112	C2 N = 172	C3 N = 253	*p*
Age (years. M ± SD)	52.49 ± 13.37	52.03 ± 13.35	53.62 ± 14.09	53.9 ± 12.8	0.470
Gender					0.238
Male	24 (7.8%)	16 (14.3%)	19 (11%)	24 (9.5%)	
Female	283 (92.2%)	96 (85.7%)	153 (89%)	229 (90.5)	
Environment					0.640
Urban	188 (61.2%)	75 (67%)	112 (65.1%)	156 (61.7%)	
Rural	119 (38.8%)	37 (33%)	60 (34.9%)	97 (38.3%)	
Hospital stay (days. M ± SD)	4.62 ± 1.87	4.42 ± 1.75	3.95 ± 2.40	4.10 ± 1.88	0.001 *
Preoperative stay (days. M ± SD)	1.67 ± 1.28	1.82 ± 1.24	1.37 ± 0.77	1.51 ± 1.37	0.007 *
Postoperative stay (days. M ± SD)	2.94 ± 1.33	2.6 ± 1.15	2.58 ± 2.23	2.59 ± 1.16	0.012 *
Duration of surgery (minutes. M ± SD)	133.62 ± 44.47	139.49 ± 52.72	153.41 ± 58.07	162.63 ± 57.02	<0.001 *
Type of surgery					0.699
Total Thyroidectomy	242 (78.8%)	89 (79.5%)	140 (81.4%)	209 (82.6%)	
Lobectomy	65 (21.2%)	23 (20.5%)	32 (18.6%)	44 (17.4%)	
Tumor size (cm. M ± SD)	1.65 ± 0.91	2.20 ± 1.72	1.69 ± 0.93	1.85 ± 0.96	<0.001 *

* *p* value lower than 0.05.

**Table 4 cancers-15-03032-t004:** Clinical and pathological characteristics of papillary thyroid cancer.

Variables	Pre-COVID-19 N = 125	C1 N = 54	C2 N = 79	C3 N = 146	*p*
Age (years. M ± SD)	53.73 ± 12.40	53.46 ± 14.58	52.67 ± 12.87	56.14 ± 13.14	0.208
Gender					0.760
Male	18 (14.4%)	8 (14.8%)	8 (10.1%)	22 (15.1%)	
Female	107 (85.6%)	46 (85.2%)	71 (89.9%)	124 (84.9%)	
Environment					0.452
Urban	82 (65.6%)	34 (63%)	48 (60.8%)	82 (56.2%)	
Rural	43 (34.4%)	20 (37%)	31 (39.2%)	64 (43.8%)	
Hospital stay (days. M ± SD)	5.14 ± 3.15	4.69 ± 2.13	4.19 ± 2.11	3.91 ± 1.48	<0.001 *
Preoperative stay (days. M ± SD)	2.0 ± 1.86	1.87 ± 1.30	1.61 ± 1.29	1.39 ± 0.96	0.003 *
Postoperative stay (days. M ± SD)	3.14 ± 2.09	2.81 ± 1.51	2.57 ± 1.26	2.52 ± 1.15	0.007 *
Duration of surgery (minutes. M ± SD)	133.15 ± 46.85	130.0 ± 44.67	155.24 ± 62.96	160.72 ± 71.21	<0.001 *
Tumor size (cm)					0.870
≤1 cm	87 (70.2%)	37 (68.5%)	51 (64.6%)	100 (68.5%)	
>1 cm	37 (29.8%)	17 (31.5%)	28 (35.4%)	46 (31.5%)	
Multifocality					0.205
Single	76 (60.8%)	31 (57.4%)	39 (49.4%)	93 (63.7%)	
Multiple	49 (39.2%)	23 (42.6%)	40 (50.6%)	53 (36.3%)	
Capsular invasion					0.127
Absent	71 (56.8%)	35 (64.8%)	36 (45.6%)	86 (58.9%)	
Present	54 (43.2%)	19 (35.2%)	43 (54.4%)	60 (41.1%)	
Vascular invasion					0.193
Absent	98 (78.4%)	46 (85.2%)	55 (69.6%)	114 (78.1%)	
Present	27 (21.6%)	8 (14.8%)	24 (30.4%)	32 (21.9%)	
pT					0.426
1	93 (77.5%)	48 (88.9%)	70 (88.6%)	113 (78.5%)	
2	16 (13.3%)	4 (7.4%)	7 (8.9%)	18 (12.5%)	
3	11 (9.2%)	2 (3.7%)	2 (2.5%)	12 (8.3%)	
4	0 (0%)	0 (0%)	0 (0%)	1 (0.7%)	
pN					0.282
0	104 (86.7%)	54 (100%)	69 (87.3%)	125 (86.8%)	
x	7 (5.8%)	0 (0%)	4 (5.1%)	5 (3.5%)	
1a	6 (5.0%)	0 (0%)	5 (6.3%)	12 (8.3%)	
1b	3 (2.5%)	0 (0%)	1 (1.3%)	2 (1.4%)	

* *p* value lower than 0.05.

## Data Availability

The datasets used and/or analyzed during the current study are available from the corresponding author on reasonable request.

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
