# Peer review of "The Changing Landscape of Thyroid Surgery during the COVID-19 Pandemic: A Four-Year Analysis in a University Hospital in Romania"

_cancers, 2023, doi:10.3390/cancers15113032_

Round 1

Reviewer 1 Report

I would like to congratulate the authors on their fascinating work regarding this interesting manuscript on Thyroid Surgery during the COVID-19 Pandemic. The manuscript is well-written and the incorporated tables make the study easy to follow.

I strongly recommend acceptance for publication of the paper after major revision.

1) I would like a brief discussion on the Bethesda classification system for reporting thyroid cytopathology  ( especially for type II and III) and consider citing the recently published articles on Bethesda II and III:

https://pubmed.ncbi.nlm.nih.gov/33749812/

Which is the percentage of incidental malignancy according to these studies for Bethesda II and III?

3) According to the literature, there is controversy on the selection of the best surgical treatment for differentiated thyroid cancer (TC), total thyroidectomy (TT), and subtotal thyroidectomy (STT). Is there an increased risk of early complications after TT in comparison with STT?

I would like a brief discussion on that and consider citing the related published article.

Author Response

Thank you very much for your review and feedback. Your comments and suggestions were on point, and we have endeavored to address and incorporate the necessary modifications, taking into account the aspects mentioned.

Comment 1&2: I would like a brief discussion on the Bethesda classification system for reporting thyroid cytopathology  ( especially for type II and III).

Which is the percentage of incidental malignancy according to these studies for Bethesda II and III?

Answer 1: Thank you very much for the refferences provided, it has helpled us indeed in searching for the answer. We tried to make as brief a discussion as possbile and  we think that :

The results of thyroid fine needle aspiration (FNA) are classified and reported using The Bethesda System for Reporting Thyroid Cytopathology (TBSRTC). Although this study has its limitations and does not provide data on this aspect, it is worth noting that this system is internationally used in the management of patients with thyroid nodular pathology. The TBSRTC comprises six categories, and the risk of incidental malignancy increases from category I to category VI. In the literature, information regarding category III varies. While the risk of incidental malignancy in category II is generally low, ranging from 0% to 3%, it varies between 5% and 15% in category III, with some studies reporting percentages as high as 26.6% to 37.8%. As a result, these patients may have a higher risk of malignancy than traditionally believed.

Comment 3: According to the literature, there is controversy on the selection of the best surgical treatment for differentiated thyroid cancer (TC), total thyroidectomy (TT), and subtotal thyroidectomy (STT). Is there an increased risk of early complications after TT in comparison with STT?

Answer 3: Indeed, the raised issue is a current topic of interest, as there are controversies surrounding the therapeutic options at present. 

The proportion of patients who underwent total thyroidectomy (TT) for differentiated thyroid cancer (TC) is significantly higher than the ones with subtotal thyroidectomy (STT). According to the available literature, there is ongoing controversy regarding the optimal surgical treatment between the two. The debate mainly arises when considering young patients with small tumors confined to one lobe of the thyroid. Studies have shown that total thyroidectomy with TSH suppression significantly reduces recurrence rates and enables accurate staging, while also facilitating subsequent radioactive iodine ablation .

Furthermore, a study involving 306 patients reported that postoperative hematoma occurred in 1.78% of cases in the TT group and 2.17% of cases in the STT group, with no statistically significant differences between the two groups . It is worth noting that temporary recurrent laryngeal nerve palsy occurred in 15.89% of patients undergoing TT and only 11.96% of patients undergoing STT, but these differences were not statistically significant.

We hope that we have successfully addressed the questions that have been raised.

Kind regards,

Dr. Calin Muntean

Reviewer 2 Report

The article is interesting and well written. I think it can be published in the present form.

The aim of this study was to highlight the changes in the surgical treatment of patients with thyroid pathology over a 4-year period.. There are other articles on this topic but it is relevant and interesting for the results reported. It is important because the findings confirm the modification of clinical and therapeutic management of patients who underwent thyroid surgery over the past 4 years, with the pandemic generating an impact whose full consequences are not yet fully known. Informations about patients can be improved. The conclusions are correcto and consistend with the study. The references are appropriate.

Tables and figures are good.

English is clear and needs only minor editing

Author Response

Thank you very much for your review and appreciation. Indeed, the COVID-19 pandemic has brought significant changes in the case of these patients. It would have certainly been interesting to analyze additional patient data, but we believe that we have taken into cosideration various aspects that present the consequences of this period on thyroid surgery.

Kind regards,

Dr. Calin Muntean

Round 2

Reviewer 1 Report

All requested changes were addressed accordingly. The manuscript is well-written and can be accepted for publication without further changes.